# Full Rescue of F508del-CFTR Processing and Function by CFTR Modulators Can Be Achieved by Removal of Two Regulatory Regions

**DOI:** 10.3390/ijms21124524

**Published:** 2020-06-25

**Authors:** Inna Uliyakina, Hugo M. Botelho, Ana C. da Paula, Sara Afonso, Miguel J. Lobo, Verónica Felício, Carlos M. Farinha, Margarida D. Amaral

**Affiliations:** BioISI—Biosystems & Integrative Sciences Institute, Faculty of Sciences, University of Lisboa, 1749-016 Lisboa, Portugal; iiuliyakina@fc.ul.pt (I.U.); hmbotelho@fc.ul.pt (H.M.B.); acdapaula@fc.ul.pt (A.C.d.P.); scafonso@fc.ul.pt (S.A.); mglobo@fc.ul.pt (M.J.L.); vmfelicio@fc.ul.pt (V.F.); cmfarinha@fc.ul.pt (C.M.F.)

**Keywords:** ABC transporters, drug action, regulatory extension, regulatory insertion, mechanism of action

## Abstract

Cystic Fibrosis (CF) is caused by mutations in the CF Transmembrane conductance Regulator (CFTR), the only ATP-binding cassette (ABC) transporter functioning as a channel. Unique to CFTR is a regulatory domain which includes a highly conformationally dynamic region—the regulatory extension (RE). The first nucleotide-binding domain of CFTR contains another dynamic region—regulatory insertion (RI). Removal of RI rescues the trafficking defect of CFTR with F508del, the most common CF-causing mutation. Here we aimed to assess the impact of RE removal (with/without RI or genetic revertants) on F508del-CFTR trafficking and how CFTR modulator drugs VX-809/lumacaftor and VX-770/ivacaftor rescue these variants. We generated cell lines expressing ΔRE and ΔRI CFTR (with/without genetic revertants) and assessed CFTR expression, stability, plasma membrane levels, and channel activity. Our data demonstrated that ΔRI significantly enhanced rescue of F508del-CFTR by VX-809. While the presence of the RI seems to be precluding full rescue of F508del-CFTR processing by VX-809, this region appears essential to rescue its function by VX-770, suggesting some contradictory role in rescue of F508del-CFTR by these two modulators. This negative impact of RI removal on VX-770-stimulated currents on F508del-CFTR can be compensated by deletion of the RE which also leads to the stabilization of this mutant. Despite both regions being conformationally dynamic, RI precludes F508del-CFTR processing while RE affects mostly its stability and channel opening.

## 1. Introduction

Cystic Fibrosis (CF), a life-threatening recessive disorder affecting ~80,000 individuals worldwide, is caused by mutations in the gene encoding the CF transmembrane conductance regulator (CFTR) protein present at the apical membrane of epithelial cells. This is the only member of the ATP-binding cassette (ABC) transporter family functioning as a channel, more precisely a cyclic Adenosine Monophosphate (cAMP)-dependent chloride (Cl^−^)/bicarbonate (HCO_3_^−^) channel. It consists of two membrane-spanning domains (MSD1/2), two nucleotide binding domains (NBD1/2) and a cytoplasmic regulatory domain (RD), which is unique to CFTR [1]. The MSDs are linked via intra- and extra-cellular loops (ICLs, ECLs, respectively). ATP binding promoting NBD1:NBD2 dimerization preceded by phosphorylation of RD at multiple sites lead to channel gating [2]. The NBD1:NBD2 and ICL4:NBD1 interfaces were shown to represent critical folding conformational sites [3,4,5] important for the gating and maturation of the CFTR protein [5,6].

Although, >2000 CFTR gene mutations were reported (http://www.genet.sickkids.on.ca), one mutation–F508del–occurs in 85% of CF patients. This NBD1 mutant fails to traffic to the plasma membrane (PM) due to protein misfolding and retention by the endoplasmic reticulum quality control that targets it for premature proteasomal degradation. F508del-CFTR folding is a complex and inefficient process but it can be rescued, at least partially, by several treatments. These include low temperature incubation [7], genetic revertants [4,5,8,9,10,11,12,13] or pharmacological agents, like “corrector” VX-809 (lumacaftor) [14], one of the first CFTR modulator drugs to receive approval from the United States Food and Drug Administration in combination with potentiator VX-770/ivacaftor. Determining the additive/ synergistic rescue of F508del-CFTR by small molecule correctors together with other rescuing agents/revertants is very valuable to determine the mechanism of action of these CFTR modulators [5].

Comparative studies of CFTR with other ABC transporters are very powerful to understand the uniqueness of some of its regions, their influence on CFTR maturation and function as well as how they affect distinctive binding of CFTR to modulator drugs. Two such unique regions are present in NBD1 of CFTR, which are absent in NBDs of other ABC transporters (Appendix A)—the regulatory extension (RE) and regulatory insertion (RI). Both regions were described as highly conformationally dynamic [3,15] following PKA phosphorylation at certain residues (^660^Ser, ^670^Ser, and ^422^Ser) [16,17]. Importantly, removal of the 32-amino acid RI (ΔRI) was reported to rescue traffic of F508del-CFTR [13].

Our first goal was to assess the impact of removing the 30 amino acid RE (ΔRE) alone or jointly with ΔRI on F508del-CFTR trafficking. Because there is some controversy regarding the RI and RE boundaries [3,13,15,18,19,20] which have structural implications, we tested the short and long versions of both regions (Appendix A). Secondly, we aimed to evaluate how RE and or RI removal from F508del-CFTR influences the rescue of this mutant by genetic revertants. Our third and final goal was to determine how the traffic and function of these combined variants of F508del-CFTR (ΔRE, ΔRI plus genetic revertants) are rescued by CFTR modulator drugs VX-809 (corrector) and VX-770 (potentiator) to gain further insight into their mechanism of action.

Our data show that although F508del-CFTR without RE did not traffic to the PM, it showed a dramatic stabilization of its immature form (evidencing a turnover rate ~2× lower than that of wt-CFTR. Results also show that, while ΔRI further increased processing of F508del-CFTR with revertants to almost wt-CFTR levels, RE removal completely abolished their processing, thus highlighting the different impact of the two dynamic regions on revertant-rescued F508del-CFTR. Most strikingly, although VX-809 rescued ΔRI-F508del-CFTR and ΔRE-ΔRI-F508del-CFTR processing to wt-CFTR levels, to achieve maximal function of F508del-CFTR, removal of just RI was insufficient, as both RI and RE had to be absent from F508del-CFTR. These data indicate that removal of these two regions has a positive effect on the rescuing efficacy of F508del-CFTR by CFTR modulators.

## 2. Results

### 2.1. Removal of Short Regulatory Extension (RE_S_) Alone or with Regulatory Insertion (RI) has No Impact on 508del-CFTR Processing

Given the controversy in defining both RI and RE, we chose to generate two versions of these regions because of the respective structural implications. Indeed, RI_L_ and RE_L_ are the complete regions which are absent in other ABC transporters (Appendix A), whereas, their shorter versions appeared as structurally meaningful to be removed: RI_S_ is strictly the region described as destructured in the crystal structure and RE_S_ is the core of RE, i.e., just the β-strand [19]. The impact of deleting RE_S_—short RE (Δ^654^Ser-Gly^673^, Appendix A)—was first assessed, either alone or together with RI in its short and long variants (ΔRI_S_ and ΔRI_L_), on the in vivo processing of wt- and F508del-CFTR by Western blot (WB) of Baby Hamster Kidney (BHK) cells stably expressing such variants (Figure 1B,E,F and Table 1). Processing of CFTR was assessed by WB in terms of its fully-glycosylated form (also termed band C) corresponding to post-Golgi forms, assumedly at the PM. Unprocessed CFTR, in turn corresponds to its core-glycosylated, ER-specific form (also termed band B). Removal of RE_S_ had no impact on processing of either F508del-CFTR or wt-CFTR (Figure 1B, lanes 5,10; Table 1), despite that the immature form of ΔRE_S_-F508del-CFTR appeared consistently at increased levels vs. those of F508del-CFTR (Figure 1B, lanes 5,2).

Removal of RE_S_ together with RI_L_—long RI (Δ^404^Gly-Leu^435^, Appendix A) led to an increase in processing of F508del-CFTR from 3 ± 2% to 71 ± 3% (vs. wt-CFTR levels), similarly to what had been previously reported by Alexandrov et al. for ΔRI_L_ alone [13]. In contrast, removal of RI_S_—short RI (Δ^412^Ala-Leu^428^, Appendix A) had no impact on ΔRE_S_-F508del-CFTR processing (Figure 1B, lanes 6,7; Figure 1F; Table 1). So in summary, removal of RE_S_ from the ΔRI_S_- or ΔRI_L_-F508del-CFTR variants had no further effect on their processing, despite that processing of ΔRE_S_-F508del-CFTR is different upon removal of RI_S_ or RI_L_, but this difference remains equivalent to removal RI_S_ or RI_L_ alone. Despite this lack of impact on processing, removal of RE_S_ does affect total protein expression levels of ΔRI_S_- or ΔRI_L_-F508del-CFTR variants leading to a decrease in RI_S_-F508del-CFTR (Figure 1B, lanes 3,6) and an increase in RI_L_-F508del-CFTR (Figure 1B, lanes 4,7). Interestingly, when RE_S_ and RI_S_, were jointly removed from wt-CFTR, its processing was significantly reduced to 54 ± 9%, while ΔRE_S_ jointly with ΔRI_L_ caused no impact (Figure 1B, lanes11,12, respectively; Figure 1E; Table 1). Again, these data were equivalent to removal of RI_S_ or RI_L_ alone on wt-CFTR processing (Figure 1B, lanes 8,9; Figure 1F; Table 1).

The differential effect caused by removal of RI_S_ vs RI_L_ on F508del- and wt-CFTR emphasize the importance of those 8 N-term (^404^Gly-Lys^411^) and 7 C-term (Phe^429^-Leu^435^) amino acid residues that differ between the two RI regions for the folding and processing of CFTR.

Overall, removal of RE_S_ alone or with RI has no impact on F508del-CFTR processing efficiency. 

### 2.2. Simultaneous Removal of Long Regulatory Extension (RE_L_) and Helix H9 Significantly Reduces wt-CFTR Processing but Increases Levels of F508del-CFTR Immature Form

Next, we assessed the impact of removing the long version of RE–RE_L_ (Δ^647^Cys-Ser^678^, Figure 2)—which significantly decreased wt-CFTR processing to 88 ± 5% (Figure 2A, lane 3; Figure 2C) but had no impact on F508del-CFTR (Figure 2A, lane 4; Figure 2C). As helix H9 (Δ^637^Gln-Gly^646^, Appendix A) can be considered to be part of this longer RE region [18] since it interacts with RE (Figure 1A), we also tested the effect of removing helix H9 jointly with ΔRE_L_. Deletion of both H9 (ΔH9) and RE_L_ further decreased the residual processing of ΔRE_L_-F508del-CFTR from 9 ± 2% to 0±0% and in fact the same happened for ΔH9 alone (Figure 2A, lanes 8,6, respectively). Curiously, however, levels of immature F508del-CFTR were significantly increased when both RE_L_ and H9 were removed (Figure 2A, lane 8), similarly to ΔRE_S_-F508del-CFTR (Figure 1B) and in contrast to ΔRE_L_-F508del-CFTR.

For wt-CFTR, ΔRE_L_-ΔH9 also led to a decrease in processing (55 ± 3%), which was more pronounced than for ΔRE_L_ alone, but interestingly removal of H9 helix alone only reduced processing to 72 ± 4% (Figure 2A, lanes 7,5, respectively).

Simultaneous removal of RE_L_ and helix H9 significantly reduces wt-CFTR processing but increases levels of F508del-CFTR immature form.

### 2.3. ΔRI_S_, but Not ΔRE_S_, Abolishes the Plasma Membrane Rescue of F508del-CFTR by Revertants

In order to test whether the stabilized immature form of ΔRE_S_-F508del-CFTR could be rescued to mature form, we next tested the effects of removing the RE_S_ from F508del-CFTR with R1070W and G550E revertants. Indeed, both the R1070W and G550E revertants rescued ΔRE_S_-F508del-CFTR to 33 ± 7% and 37 ± 5%, respectively (Figure 1D, lanes 6,10; Table 1).

However, removal of RI_S_ from F508del-CFTR with R1070W and G550E revertants, virtually abolished the rescue of F508del-CFTR by both revertants (Figure 1D, lanes 4,8; Table 1). Moreover, processing of ΔRI_L_-F508del-CFTR was further increased by either R1070W or G550E up to 96 ± 2% and 92 ± 4%, respectively (Figure 1D, lanes 5,9; Table 1). Alone, R1070W and G550E rescued F508del-CFTR from 8 ± 1% to 34 ± 3% and 42 ± 4%, respectively (Figure 1D, lanes 3,7; Table 1), as we previously reported [5].

Interestingly, the impaired processing of ΔRI_S_-wt-CFTR (44 ± 2%) was significantly rescued by G550E (to 82 ± 3%), but curiously it was further reduced by R1070W (to 11%) i.e., close to levels of F508del-CFTR processing (Figure 1C, lanes 8,4; Table 1). Of note that R1070W alone also reduced wt-CFTR processing to 69 ± 7%, but this reduction could be compensated by the removal of RI_L_ (92 ± 2%) (Figure 1C, lanes 3,5; Table 1), while G550E alone caused no effect on wt-CFTR processing (Figure 1C, lane 7).

In summary, ΔRI_S_, but not ΔRE_S_, abolishes the plasma membrane rescue of F508del-CFTR by genetic revertants R1070W and G550E.

### 2.4. ΔRI_L_ Synergises with VX-809, but Not with Revertants to Rescue ΔRE_S_-F508del-CFTR Processing

To further test how the stabilized immature form of ΔRE_S_-F508del-CFTR could be pharmacologically rescued, we then assessed the impact of corrector VX-809 (which per se promotes maturation of the F508del-CFTR) on the processing of this and of the other variants, to obtain structural insight on the effects of this novel drug. Although ΔRE_S_-F508del-CFTR could not be rescued by VX-809, data show that this small molecule was able to rescue ΔRI_S_-ΔRE_S_-F508del-CFTR (from 7 ± 2% to 20 ± 4%) and further increased processing of ΔRI_L_-ΔRE_S_-F508del-CFTR to wt-CFTR levels (from 71 ± 3% to 96 ± 2%) (Figure 3A, lanes 6,7; Figure 3E; Table 1). These data suggest a strong synergistic effect between VX-809 and ΔRI_L_ (and with ΔRI_S_, albeit to a lesser extent) to rescue ΔRE_S_-F508del-CFTR processing. This is particularly interesting because VX-809 did not rescue processing of the ΔRE_S_-, nor ΔRI_S_-F508del-CFTR variants (Figure 3A, lanes 5,3; Table 1) while it recovered the processing of ΔRI_L_-F508del-CFTR to wt-CFTR levels (Figure 3A, lane4; Table 1). Notably, VX-809 was able to almost completely revert processing impairment of ΔRI_S_-wt-CFTR and ΔRI_S_-ΔRE_S_-CFTR (from 44 ± 2% and 54 ± 9%) to 90 ± 3% and 93 ± 3%, respectively (Figure 3A, lanes 8,11; Figure 3D; Table 1).

Strikingly, VX-809 had no effect on processing of ΔRE_S_-F508del-CFTR with R1070W or G550E (Figure 3C, lanes 6,10; Figure 3E; Table 1), while it further rescued processing of F508del-CFTR variants with R1070W or G550E alone to 46 ± 4% and 73 ± 6%, respectively (Figure 3C, lanes 3,7; Figure 3E; Table 1), as reported [5]. Similarly, VX-809 caused no significant further rescue on the processing of ΔRI_L_-F508del-CFTR with those revertants, but these variants already had processing levels close to those of wt-CFTR (Figure 3C, lanes 5,9; Figure 3E; Table 1). Interestingly, the impaired processing of ΔRE_S_-R1070W-wt-CFTR (71 ± 6%) was also reverted by VX-809 to 90 ± 3% (Figure 3B, lane 6; Table 1).

Corrector VX-809 failed to rescue any of the ΔRE_L_-, ΔH9-, and ΔRE_L_-ΔH9-F508del-CFTR variants (Figure 2B,C), while significantly increasing the processing of the ΔRE_L_-ΔH9-wt-CFTR from 55 ± 3% to 69 ± 3% (Figure 2B, lane 7; Figure 2C).

Altogether, these results suggest that RI_L_ and the genetic revertants interfere with the rescue of F508del-CFTR by VX-809.

### 2.5. ΔRI_L_-ΔRE_S_-F508del-CFTR Levels at the Plasma Membrane are Equivalent to Those of wt-CFTR

To determine the fraction of the above CFTR variants that localize to the PM, we used quantitative cell surface biotinylation. These data showed that PM levels of ΔRI_L_-ΔRE_S_-F508del-CFTR were equivalent to those of wt-CFTR, while those of ΔRI_L_-F508del-CFTR were significantly lower (Figure 4A, lanes 5,4; Figure 4B). Data also confirmed that ΔRE_S_ did not induce appearance of F508del-CFTR at the cell surface (data not shown). Corrector VX-809 further increased the PM expression of ΔRI_L_-ΔRE_S_-F508del-CFTR to levels that are significantly higher than those of wt-CFTR (Figure 4A, lanes 2,9, Figure 4B). This compound also significantly increased PM levels of ΔRI_L_-F508del-CFTR to similar levels of wt-CFTR (Figure 4A, lanes 2,8; Figure 4B).

Given the very significant stabilization of immature ΔRE_S_-F508del-CFTR (Figure 1B), next, we determined how removal of RE_S_ affected the processing efficiency and the turnover of the F508del- and ΔRI_L_-F508del-CFTR variants. To this end, we performed pulse-chase experiments (Figure 4C,D) and indeed our results revealed that ΔRE_S_ very significantly stabilized immature F508del-CFTR not just relatively to F508del-CFTR but to levels even significantly higher than those of wt-CFTR (Figure 4C,D). Indeed, our data show that although F508del-CFTR without RE did not traffic to the PM, it showed a dramatic stabilization of its immature form (evidencing a turnover rate ~2× lower than that of wt-CFTR. It should be noted that the turnover of F508del-CFTR itself is ~1.4×faster vs. wt-CFTR, thus this stabilization (of ~3× vs. F508del-CFTR) represents indeed a massive stabilization. Interestingly, this stabilizing effect was no longer significant for ΔRI_L_-ΔRE_S_-F508del-CFTR nor for ΔRI_L_-F508del-CFTR, the latter being equivalent to wt-CFTR (Figure 4C,D). Removal of RI_S_ from F508del-CFTR did not stabilize its turnover (data not shown). As to removal of either RE_S_ or RI_S_ from wt-CFTR, it did not affect the processing efficiency or the turnover vs. wt-CFTR (Appendix A).

In summary, levels of ΔRI_L_-ΔRE_S_-F508del-CFTR at the PM are equivalent to those of wt-CFTR.

### 2.6. VX-809 Jointly with RE and RI Removal Completely Restored F508del-CFTR Function as Chloride Channel

To investigate the channel function of the ΔRE_S_- and ΔRI_L_-F508del-CFTR variants, we used the iodide efflux technique. Removal of RE_S_ alone had no impact on F508del-CFTR function (Figure 5A; black dash line in Appendix A) as expected from the lack of processed form for this variant (Figure 1B). However, when RE_S_ and RI_L_ were removed jointly from F508del-CFTR, functional levels reached 78 ± 7% of wt-CFTR (Figure 5A,C; black dotted line in Appendix A). As to removal of RI_L_ alone from F508del-CFTR, it restored function to 92 ± 7% of wt-CFTR (Figure 5A,C; black line in Appendix A), as described [13]. Also, the delay in peak response observed for F508del-CFTR (min = 4) vs. wt-CFTR (min = 2), was partially corrected (to min = 3) both by removal of RI_L_ alone from F508del-CFTR or together with RE_S_ (Figure 5C; Appendix A, black line and dotted line).

Introduction of G550E and R1070W revertants into the ΔRI_L_-F508del-CFTR variant slightly but not significantly decreased its function from 92 ± 7% to 71 ± 8% and 76 ± 8%, respectively (Figure 5A,C; black lines in Appendix A), in parallel to the respective processing levels (Figure 1). In fact, these variants showed higher activity levels than the revertants containing RI_L_ (29 ± 3% and 56 ± 6%, respectively). Interestingly however, both G550E- and R1070W-ΔRI_L_-F508del-CFTR fully corrected (to min = 2) the delay in peak response of F508del-CFTR, which was still partially present in ΔRI_L_- and ΔRI_L_-RE_S_-F508del-CFTR (min = 3).

The effect of VX-809 (3µM, 48 h at 37 °C) was also examined at functional level on the ΔRE_S_-, ΔRI_L_-ΔRE_S_- and ΔRI_L_-F508del-CFTR variants. VX-809 shows a tendency to increase in ΔRE_S_-ΔRI_L_-F508del-CFTR function (84 ± 8% of wt-CFTR) which was parallel to the observed increase in PM expression (Figure 4; Figure 5A,C; black dotted line in Appendix A). Also, ΔRE_S_-ΔRI_L_-F508del-CFTR fully corrected (to min = 2) the delay in peak response of F508del-CFTR which was still partially present in ΔRI_L_- and ΔRI_L_-ΔRE_S_-F508del-CFTR (min = 3).

In contrast to the observed increase in processing, VX-809 caused no significant increase in the function of ΔRI_L_-F508del-CFTR and actually a slight, but not significant decrease was observed and there was still the same delay in peak response (Figure 5A,C; black line in Appendix A).

A similar slight decrease in function was observed for VX-809 on the ΔRI_L_-R1070W-F508del-CFTR variant (Figure 5A,C; black line in Appendix A), but the most striking result was the significant decrease caused by VX-809 on function of ΔRI_L_-G550E-F508del from 76 ± 8% to 41 ± 3% (Figure 5A; black line in Appendix A). Nevertheless, both revertant variants of ΔRI_L_-F508del-CFTR showed no change in peak response which was still the same as wt-CFTR (min = 2).

Overall, VX-809 jointly with RE and RI removal completely restored F508del-CFTR function as chloride channel.

### 2.7. VX-770-Stimulated Currents of CFTR Variants are Dramatically Decreased by ΔRI_L_ but not by ΔRI_L_-ΔRE_S_

Given the interesting and diverse results observed for the ΔRE_S_ and ΔRI_L_ under VX-809, next we tested the effects of potentiator VX-770 (Ivacaftor), an approved drug for CF patients with gating mutations (Figure 5B) on these variants and in combination with VX-809, for F508del/F508del patients. Most strikingly, our data show that VX-770/Forskolin (Fsk) significantly decreased the function of ΔRI_L_-F508del-CFTRin comparison with potentiation by Gen/Fsk (Figure 5B,C) from 92 ± 7% to 58 ± 7% of wt-CFTR (Figure 5B,C; black line in Appendix A). As ΔRE_S_-F508del-CFTR was not processed, we did not test the effect of VX-770 on this variant, but when the ΔRE_S_-ΔRI_L_-F508del-CFTR variant was assessed for its function after acute application of VX-770 with Fsk, a tendency for increase in its function (to 87 ± 3% of wt-CFTR) was observed *versus* its function under Gen/Fsk (78 ± 7% of wt-CFTR) (Figure 5B,C; black dotted line in Appendix A).

The most dramatic result, however, was observed for the ΔRI_L_-G550E-F508del-CFTR variant which under VX-770/Fsk only had 4 ± 1% function of wt-CFTR, while under Gen/Fsk it had 76 ± 8% (Figure 5B,C; black line in Appendix A). For ΔRI_L_-R1070W-F508del-CFTR this decrease was not observed (Figure 5B,C; black line in Appendix A), and in fact this variant had a slightly higher function under VX-770 stimulation (72 ± 4%) than ΔRI_L_-F508del-CFTR (58 ± 7%) (Figure 5B,C, black line in Appendix A). These results suggest that somehow removal of RI_L_ can increase the chances of VX-770 blocking the channel pore.

The delay in peak response of F508del- comparing with wt-CFTR was not corrected for ΔRI_L_-F508del-CFTR (min = 3), but it was corrected for ΔRI_L_-ΔRE_S_-F508del-CFTR (min = 2) (black and dotted black lines in Appendix A, respectively). Interestingly, on the other hand ΔRI_L_-R1070W-F508del-CFTR showed the quickest peak response (at min = 1), in contrast to either R1070W-F508del-CFTR or ΔRI_L_-F508del-CFTR, both at min = 3.

Similarly, VX-770 also significantly decreased the function of ΔRI_L_-wt-CFTR from 127 ± 15% under Gen to 57 ± 6% (Appendix A).

These results imply that, VX-770-stimulated currents of CFTR variants are dramatically decreased by ΔRI_L_ but not by ΔRI_L_-ΔRE_S._

## 3. Discussion

The main goal of this study was to understand how the removal of the regulatory extension (ΔRE) alone or with the regulatory insertion (ΔRI)—two highly conformationally dynamic regions—impact the rescue of F508del-CFTR processing and function by two compounds—VX-809 and VX-770—which in combination were recently approved to clinically treat F508del-homozygous patients. Indeed, CFTR is the sole ABC transporter that functions as a channel and thus, these highly dynamic RI and RE regions which are absent in other ABC transporters, may be of high relevance to understand how CFTR differs from other ABCs, namely in its function as a channel.

These regions RI and RE were originally suggested to be positioned to impede formation of the NBD1-NBD2 dimer required for channel gating [15,19]. Indeed, both RI and RE were shown to be mobile elements in solution that bind transiently to the core of NBD in the β-sheet and α/β subdomains of NBD1 [3]. Similarly to what demonstrated for RI [13], it is plausible to posit that the dynamic flexibility of the RE may also result in exposure of hydrophobic surfaces thus contributing to the dynamic instability of NBD1 and thus contribute to the low folding efficiency of F508del-CFTR. The RE is a ~30-residue segment at NBD1 C-terminus, so-called because it goes beyond canonical ABC NBDs [15]. Although this region was absent from the solved CFTR-NBD1 crystal structure [19], it was described as a helix packing against NBD1 at the NBD1:NBD2 interface and NMR data showed it has significant conformational flexibility [3,21]. Notwithstanding, there is some controversy regarding the RE boundaries. The RE (previously called H9c) was defined as ^655^Ala-Ser^670^ [15,19], ^639^Asp-Ser^670 20^, or ^654^Ser-Gly^673^ [3]. According to the crystal structure (PDB ID 2PZE), NBD1 extends only to ^646^Gly, where RD begins, so RE could be proposed to start at ^647^Cys [3,18], thus being considered as part of RD, the latter absent in other ABC transporters [1]. As for the RI (previously called S1-S2 loop) its limits are also variable, being firstly described as a ~35-residue segment (^405^Phe-^436^Leu) consisting of α-helices H1b and H1c [15] (Appendix A). Later, however, these limits were proposed to be ^404^Gly-Leu^435^ [13,20] or ^405^Phe-Leu^436^ [18].

### 3.1. Impact of RE and RI on CFTR Processing and Function

Our data shown here on CFTR variants depleted of different versions of the RE dynamic region demonstrate that unlike RI deletion, removal of short RE—ΔRE_S_ (Δ^654^Ser-Gly^673^) did not per se rescue F508del-CFTR processing. Nevertheless, ΔRE_S_ dramatically stabilized the immature form of F508del-CFTR (see Figure 6). In fact, our pulse-chase experiments show that the immature form of ΔRE_S_-F508del-CFTR exhibits a turnover rate which is ~2-fold lower than that of wt-CFTR. Of note, when Aleksandrov et al. removed the dynamic region RI from F508del-CFTR they found a dramatic increase in the channel thermostability, even augmented for higher temperatures [13]. Interestingly however, while RE_S_ removal from ΔRI_L_-F508del-CFTR did not significantly affect processing (71% vs. 78% for ΔRE_S_-ΔRI_L_-F508del-CFTR and ΔRI_L_-F508del-CFTR, respectively, in Table 1, it further reduced its function from 92 to 78% (ΔRI_L_-F508del-CFTR vs. ΔRE_S_-ΔRI_L_-F508del-CFTR, respectively, Figure 5C).

The latter findings on function of ΔRE_S_-wt-CFTR and ΔRE_S_-ΔRI_L_-F508del-CFTR were somewhat surprising since the RE was previously described to impede putative NBD1:NBD2 dimerization that is required for channel gating [20]. Accordingly, an increase in CFTR activity would be expected upon ΔRE_S_ removal, which is actually the opposite of what we observe for wt-CFTR. Besides the fact that those authors studied a different RE version (^639^Asp-Ser^670^), this discrepancy could derive from structural constraints and lack of flexibility of the ‘single polypeptide’ protein that we used, vs. their ‘split’ CFTR channels (2 ‘halves’ of 633 and 668 amino acid residues) for which channel function was indistinguishable from wt-CFTR [20]. Moreover, since RD phosphorylation is required for channel gating [24,25], the reduced function of ΔRE_S_-wt-CFTR may also result from absence of ^660^Ser and ^670^Ser [17,21]. Although the RD contains more than ten PKA phospho-sites and no individual one is essential, phosphorylation of increasing numbers of sites enables progressively greater channel activity [26].

Removal of a longer RE version (ΔRE_L_: Δ^647^Cys-Ser^678^) was without effect on F508del-CFTR processing and significantly reduced that of wt-CFTR to 88%. Such differential impact on wt- and F508del-CFTR is consistent with the conformational heterogeneity between these two proteins lacking both RI and RE [27].

Removal of RI short version, RI_S_ (Δ^412^Ala-Leu^428^) significantly reduced wt-CFTR processing to less than half of its normal levels, while not rescuing F508del-CFTR processing, thus, being essential for CFTR proper folding. This is in contrast to removal of long RI (ΔRI_L_) which led to 78% processing F508del-CFTR, as reported [13] but without impact on wt-CFTR. These data indicate that those 8/7 amino acid residues at the N-term/ C-term of RI_S_ (and absent in RI_L_) impair the folding efficiency and processing of both wt- and F508del-CFTR. Despite the difficulty in speculating how those amino acid residues can specifically impact of F508del-CFTR folding and processing, the fact that structurally RI_S_ is strictly the region described as destructured in the crystal structure [19] and RI_L_ includes some flanking residues restricting its mobility may explain the observed difference.

Indeed, the recently published cryo-EM structures of human CFTR [22,28,29] indicate that RI_S_ is structurally disordered. Consistently, the RI loop in NBD1 is described in the zebrafish CFTR cryo-structure [30,31] to contribute to the amorphous density that is observed between NBD1 and the elbow helix of TMD2 [32]. Our data for the RI are consistent with these findings.

From a structural point of view, models of the full-length CFTR protein suggest that the two regions (RI and RE) behave differently. The distances between the extremities of the deleted parts are indeed different (ΔRI_S_ = 18Å, ΔRI_L_= 10Å versus ΔRE_S_ = 27Å, ΔRE_L_ = 30Å). The longer distances observed for ΔRE imply a substantial reorganization of the C-terminal parts of NBD1 and NBD2, which precludes a clear understanding of what might happen upon these deletions. In contrast, the lower distances observed for ΔRI (near a possible minimum of 6Å) allows a better simulation of the possible structural behavior of these constructs. Indeed, preliminary molecular dynamics simulations of the ΔRI_L_-F508del-CFTR suggested that in this case, part of the RD is reorganized and partially takes space left by the ΔRI deletion, thereby substituting it for tight contacts with NBD2. Meanwhile, following a probable allosteric effect [13,18,32], contact of NBD1-F508 with ICL4 residue L1077 (a likely essential contact for channel opening), is nearly completely restored by I507. Such a feature is not observed in the F508del-CFTR model [33].

### 3.2. Effect of VX-809 on F508del-CFTR Variants Lacking RE and RI

As for rescue of CFTR variants lacking RE and RI by CFTR modulators, VX-809 restored the processing of both ΔRE_S_-ΔRI_L_- and ΔRI_L_-F508del-CFTR variants equally well and to wt-CFTR levels (from 71-78% to 92-96%. These data suggest a strong synergistic effect between VX-809 and ΔRI_L_ to rescue ΔRE_S_-F508del-CFTR processing and thus some possible interference of the regulatory insertion with VX-809 binding to F508del-CFTR (see Figure 6). Indeed, although previous studies [34] suggested putative binding of VX-809 to NBD1:MSD2 (ICL4) interface (see Figure 6) or to TMD1 [35], they also suggested possibility of further F508del-CFTR correction at distinct conformational sites, so data shown here suggest that VX-809 may also bind to the regulatory insertion.

Interestingly, the processing defect of ΔRI_S_-wt-CFTR (but not of ΔRI_S_-F508del-CFTR) was rescuable by VX-809 to 90%, indicating that the amino acid stretches of the RI_L_ that remain present in RI_S_ do not affect the rescue of ΔRI_S_-wt-CFTR but preclude rescue of F508del-CFTR by VX-809.

We also tested the impact of removing helix 9 (H9) which precedes the RE (^635^Gln-Gly^646^), just after H8 (^630^Phe-Leu^634^), both helices proposed to interact with the NBD1:NBD2 heterodimer interface by folding onto the NBD1 β-subdomain [21,27] as well as to bind ICL4 near Phe508. When H9 is present, ΔRE_L_-wt-CFTR processing appears to be favored by VX-809, suggesting some synergy of this small molecule with H9 helix to correct the conformational defect(s) caused by ΔRE_L_ on wt-CFTR. These data are in contrast to ΔH9-F508del-CFTR which exhibits 0% processing with or without ΔRE_L_ and no rescue by VX-809. We can assume that H9 also contacts RE, at least in some states, such as when the RE adopts different conformations as previously suggested [19].

Most strikingly, and in contrast to its effect on processing, was the effect of ΔRI_L_ on the VX-770 stimulated currents which were decreased by almost half vs those stimulated by Gen/Fsk (92% to 58%). These data seem to indicate that the absence of the regulatory insertion could impair (and its presence favor) binding of VX-770 to CFTR (see Figure 6). Surprisingly, the removal of RE_S_ from ΔRI_L_-F508del-CFTR could correct this defect and restore the maximal function by VX-770 (see Figure 6). Indeed, under VX-770, ΔRE_S_-ΔRI_L_-F508del-CFTR exhibited significantly higher activity (87%) than ΔRI_L_-F508del-CFTR (58%). In contrast, removal of RE_S_ from wt-CFTR significantly reduced its functionin comparison with that of wt-CFTR (down to 70%) but had no effect on processing. Our study does not address the mechanism coupling the RI to F508del-CFTR pharmacological rescue, namely, why is the RI essential for rescue by VX-770 or why is it inhibitory for rescue by VX-809. Nevertheless, insight may be obtained from experimental structural data. In a recent cryo-EM structure of full-length human CFTR [29] VX-770 was found to bind in a transmembrane location coinciding with a hinge involved in gating. The authors suggested that populating this binding pocket may stabilize the intermolecular rotation opening the CFTR channel upon ATP binding at the NBD1-NBD2 interface. Given that the RI is located near the NBD interface and ATP binding site [29] we may speculate that RI deletion hampers NBD dimerization and, therefore, rescue by VX-770. Regarding VX-809, we may speculate that the RI interferes with drug binding.

### 3.3. Impact of F508del-Revertants on CFTR Variants Lacking RE and RI

Another goal of the present study was to assess how presence of the F508del-CFTR revertants G550E and R1070W [5,10,11] influence variants without RE and RI. Remarkably, our data show that the presence of either of these revertants did not affect ΔRE_S_-F508del-CFTR processing, but both of them further increased processing (but not function) of ΔRI_L_-F508del-CFTR to almost levels of wt-CFTR: 92–96% (G550E) and 71–76% (R1070W).

In contrast, removal of RI_S_ from either of these F508del-CFTR revertants completely abolished their processing, emphasizing how important the different residues between RI_S_ and RI_L_ are for F508del-CFTR conformers partially rescued by the revertants. We can speculate that RI_S_ removal from F508del-CFTR has an effect on folding equivalent to that of either G550E or R1070W.

Interestingly, regarding wt-CFTR processing, G550E (at the NBD1:NBD2 dimer interface) partially recovered the negative effect caused by RI_S_ removal, thus further suggesting that this revertant and the destructured region of RI may be allosterically coupled, since they do not plausibly interact (Appendix A). On the contrary, R1070W (at the NBD1:ICL4 interface), negatively affected processing of wt-CFTR (to 69%) and of ΔRI_S_-wt-CFTR (from 44% to 11%), while not affecting ΔRI_L_-wt-CFTR. R1070W rescues F508del-CFTR because ^1070^Trp fills the gap created by deletion of residue ^508^Phe [36]. It is not surprising that it perturbs CFTR folding due to clashing of ^508^Phe and ^1070^Trp residues. It is nevertheless, curious that R1070W affect more the processing ΔRI_S_-wt-CFTR than wt-CFTR, to levels of F508del-CFTR, suggesting that both changes affect the same region of the molecule. Rescue of both R1070W-wt-CFTR and R1070W-ΔRI_S_-wt-CFTR by VX-809 further supports this concept.

The most striking effect of ΔRI_L_ however, was the almost complete abolition of VX-770-stimulated current of G550E-F508del-CFTR to levels even lower than those observed for ΔRI_L_-F508del-CFTR (see above). It was suggested that VX-770 binds directly to CFTR to the MSDs (although it is not defined the exact binding site), in phosphorylation-dependent but ATP-independent manner and away from the canonical catalytic site [23,37]. Both RD and RI were suggested by Eckford and colleagues as putative binding regions for VX-770 [37]. Nevertheless, the pharmacological effect of VX-770 remains robust in the absence of the RD [23] and here we demonstrate that it does indeed require RI since VX-770 is unable to stimulate either ΔRI_L_-F508del-CFTR or ΔRI_L_-G550E- F508del-CFTR (see Figure 6). We can speculate that removal of RI_L_ increases the chances of VX-770 blocking the pore. In contrast, this would not occur for genistein, which has been proposed to bind to the NBDs interface [38], thus probably explaining why it does not cause this effect.

## 4. Materials and Methods

### 4.1. CFTR Variants, Cells, and Culture Conditions

Several CFTR deletion variants were produced by site-directed mutagenesis corresponding to the removal of residues: ΔRI_L_–^404^Gly-Leu^435^; ΔRI_S_–^412^Ala-Leu^428^; ΔRE_L_–^647^Cys-Ser^678^; ΔRE_S_–^654^Ser-Gly^673^; ΔH9–^637^Gln-Gly^646^; ΔRE_L_-ΔH9–^637^Gln-Ser^678^; ΔRI_S_-ΔRE_S_–both ^412^Ala-Leu^428^ and ^654^Ser-Gly^673^; and ΔRI_L_-ΔRE_S_–^404^Gly-Leu^435^ and ^654^Ser-Gly^673^. All the constructs were produced using full length wt-CFTR and F508del-CFTR.

BHK cells lines expressing ΔRI_L_-, ΔRI_L_-F508del-, ΔRI_L_-G550E-, ΔRI_L_-G550E-F508del, ΔRI_L_-R1070W-, ΔRI_L_-R1070W-F508del, ΔRI_S_-, ΔRI_S_-F508del, ΔRI_S_ -G550E-, ΔRI_S_-G550E-F508del, ΔRI_S_-R1070W-, ΔRI_S_-R1070W-F508del, ΔRE_S_-, ΔRE_S_-F508del, ΔRE_S_-G550E, ΔRE_S_-G550E-F508del, ΔRE_S_-R1070W-, ΔRE_S_-R1070W-F508del, ΔRE_L_-, ΔRE_L_-F508del, ΔRI_S_-ΔRE-, ΔRI_S_-ΔRE_S_-F508del, ΔRI_L_-ΔRE-, ΔRI_L_-ΔRE_S_-F508del, ΔH9-, ΔH9-F508del, ΔRE_L_-ΔH9-, and ΔRE_L_-ΔH9-F508del-CFTR were produced and cultured as previously described ^11^. Cells were cultured in DMEM/F-12 medium containing 5% (*v/v*) Fetal bovine serum (FBS) and 500 µM of methotrexate. For some experiments, cells were incubated with 3 µM VX-809 or with the equivalent concentration of Dimethyl sulfoxide (DMSO, Control) for 48 h at 37 °C.

### 4.2. Western Blot

To study the effect of removal of regulatory extension (RE) and or regulatory insertion (RI) in combination with genetic revertants and VX-809, cells were incubated for 48 h at 37 °C with 3 µM VX-809. After incubation, cells were lysed, and extracts analyzed by Western blot (WB) using the anti-CFTR 596 antibody (Ab) or anti-calnexin Ab as a loading control. Score corresponds to the percentage of band C to total CFTR (bands B + C) as normalized to the same ratio in samples from wt-CFTR expressing cells. Blot images were acquired using BioRad ChemiDoc XRC+ imaging system and band intensities were measured using Image Lab analysis software.

### 4.3. Pulse-Chase and Immunoprecipitation

BHK cells lines stably expressing CFTR variants were starved for 30 min in methionine-free α-modified Eagle’s medium or minimal essential medium and then pulsed for 30 min in the same medium supplemented with 100 *µ*Ci/mL [^35^S] methionine. After chasing for 0, 0.5, 1, 1.5, 2, and 3 h in a-modified Eagle’s medium with 8% (*v/v*) fetal bovine serum and 1mM non-radioactive methionine, cells were lysed in 1 mL of Radioimmunoprecipitation assay (RIPA) buffer [1% (*w/v*) deoxycholic acid, 1% (*v/v*) Triton X-100, 0.1% (*w/v*) Sodium dodecyl sulfate (SDS), 50 mM Tris, pH 7.4, and 150 mM NaCl]. The immunoprecipitation (IP) was carried out using the anti-CFTR 596 antibody in independent experiments and Protein G–agarose or Protein A–Sepharose beads. Immunoprecipitated proteins were eluted from the beads with sample buffer for 1h at room temperature and then electrophoretically separated on 7% (*w/v*) polyacrylamide gels. Gels were pre-fixed in methanol/acetic acid (30:10, *v/v*), washed in water and, for fluorography, soaked in 1M sodium salicylate for 60 min. After drying at 80 °C for 2 h, gels were exposed to X-ray films and further analyzed and quantified by densitometry.

BHK cells lines stably expressing CFTR variants were starved for 30 min in methionine-free α-modified Eagle’s medium or minimal essential medium and then pulsed for 30 min in the same medium supplemented with 100 *µ*Ci/mL [^35^S] methionine. After chasing for 0, 0.5, 1, 1.5, 2, and 3 h in a-modified Eagle’s medium with 8% (*v/v*) fetal bovine serum and 1mM non-radioactive methionine, cells were lysed in 1 mL of Radioimmunoprecipitation assay (RIPA) buffer [1% (*w/v*) deoxycholic acid, 1% (*v/v*) Triton X-100, 0.1% (*w/v*) Sodium dodecyl sulfate (SDS), 50 mM Tris, pH 7.4, and 150 mM NaCl]. The immunoprecipitation (IP) was carried out using the anti-CFTR 596 antibody in independent experiments and Protein G–agarose or Protein A–Sepharose beads. Immunoprecipitated proteins were eluted from the beads with sample buffer for 1h at room temperature and then electrophoretically separated on 7% (*w/v*) polyacrylamide gels. Gels were pre-fixed in methanol/acetic acid (30:10, *v/v*), washed in water and, for fluorography, soaked in 1M sodium salicylate for 60 min. After drying at 80 °C for 2 h, gels were exposed to X-ray films and further analyzed and quantified by densitometry.

### 4.4. Iodide Efflux

CFTR-mediated iodide effluxes were measured at room temperature using the cAMP agonist forskolin (Fsk 10 µM) and the CFTR potentiator genistein (Gen, 50 µM) or VX-770 (10 µM) and Gen (50 µM).

### 4.5. Biochemical Determination of the Plasma Membrane Levels of CFTR

To determine plasma membrane levels of CFTR protein, we performed cell surface biotinylation in BHK cells cultured on permeable growth supports or tissue culture plates using cell membrane impermeable EZ-Link™ Sulfo-NHS-SS-Biotin, followed by cell lysis in buffer containing 25 mM HEPES, pH 8.0, 1% (*v/v*) Triton, 10% glycerol (*v/v*), and Complete Protease Inhibitor Mixture, as described previously [39,40]. Biotinylated proteins were isolated by streptavidin-agarose beads, eluted into SDS-sample buffer, and separated by 7.5% (*w/v*) SDS-PAGE.

### 4.6. Multiple Sequence Alignment

Sequences for NBD domains of ABC transporters were obtained from Uniprot [41] (human and mouse CFTR) or the PDB [42] (1B0U, 1L2T, 1G29, 1G6H, and 1JJ7). Alignments were performed with Jalview [43] using the T-Coffee [44] algorithm with default parameters.

### 4.7. Data and Statistical Analyses

The data and statistical analyses used in this study comply with the recommendations on experimental design and data analysis in pharmacology [45]. Quantitative results are shown as mean ± SEM of *n* observations. To compare two sets of data, Student’s *t*-test was used, and differences considered to be significant for *p*-values ≤ 0.05. In Western blotting (Figure 1, Figure 2 and Figure 3), pulse chase (Figure 4 and Appendix A); cell surface biotinylation (Figure 4) and in iodide efflux studies (Table 1, Figure 5, Appendix A), n represents the number of experiments performed with distinct cell cultures on different days, being thus biological replicates.

### 4.8. Reagents

All reagents used here were of the highest purity grade available. Forskolin and genistein were from Sigma-Aldrich (St. Louis, MI, USA); VX-809 and VX-770 were acquired from Selleck Chemicals (Houston, TX, USA). CFTR was detected with the mouse anti-CFTR monoclonal 596Ab, which recognizes a region of NBD2 (1204–1211) from CFFT—Cystic Fibrosis Foundation Therapeutics CFF Therapeutics [46] and calnexin with rabbit polyclonal anti-calnexin Ab SPA-860, from Stressgen Biotechnologies Corporation (Victoria, BC, Canada). Other specific reagents included: X-ray films (Kodak, supplied by Sigma-Aldrich); Complete Protease Inhibitor Mixture from Roche Applied Science (Penzberg, Germany); EZ-Link™ Sulfo-NHS-SS-Biotin from Pierce Chemical Company (Rockford, IL, USA).

## 5. Conclusions

Overall, our data show that while the presence of the regulatory insertion (RI) seems to preclude full rescue of F508del-CFTR processing by VX-809, this region appears essential to rescue its function by VX-770, thus suggesting some contradictory role in rescue of F508del-CFTR by these two modulators (Figure 7). Nevertheless, this negative impact of removing RI on VX-770-stimulated currents on F508del-CFTR can be compensated by deletion of the regulatory extension which also leads to the stabilization of this mutant. We thus propose that, despite both these regions being conformationally dynamic, RI precludes F508del-CFTR processing while RE affects mostly its stability and channel opening.

## Figures and Tables

**Figure 1 ijms-21-04524-f001:**
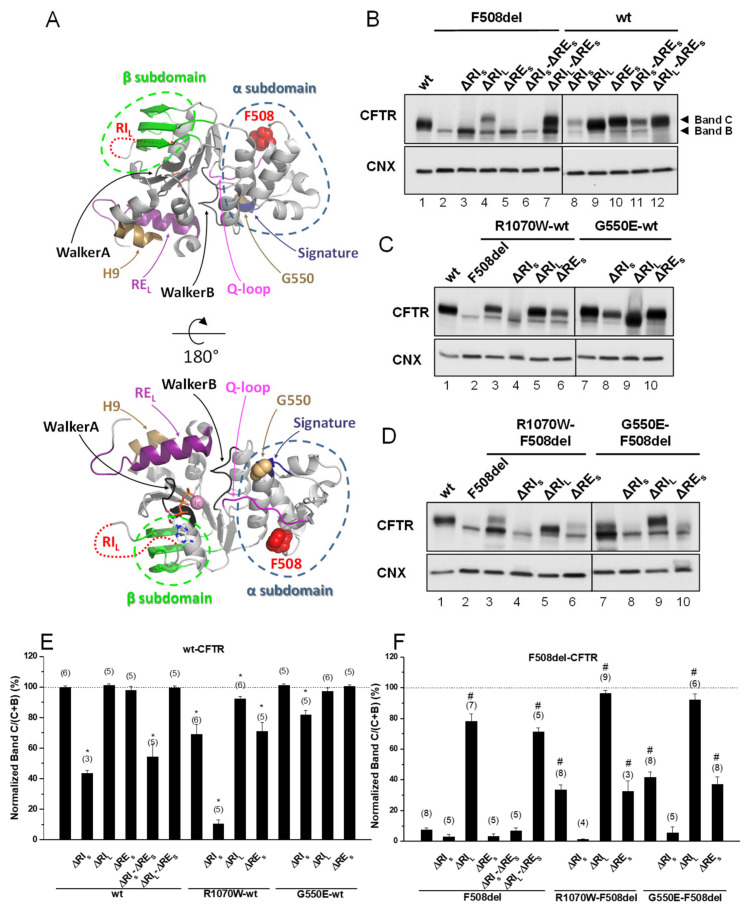
Effect of removal of short regulatory extension (ΔRE_S_) on processing of wt- and F508del-Cystic Fibrosis (CF) transmembrane conductance regulator (CFTR). (**A**) Structural features of the CFTR nucleotide binding domain 1 (NBD1) domain. Protein Data Bank identifier (PDB ID): 2BBO. The main structural regions are color coded. F508 is represented as red spheres. G550 is represented as brown spheres. ATP is shown as sticks and Mg^2+^ as a pink sphere. Regulatory insertion (RI) is unstructured in the crystal structure and represented as a dotted red line. The G550E mutation in this structure was reverted in silico. (**B**–**D**) Samples from Baby Hamster Kidney (BHK) cell lines stably expressing different CFTR variants of ΔRE_S_ and ΔRI, the latter either in its short “S” or long “L” versions (see Materials and Methods) and alone (**B**) or jointly with (**C**,**D**) genetic revertants, as indicated, were analyzed for CFTR protein expression by Western blot (WB) with anti-CFTR 596 Ab and also anti-calnexin (CNX) as loading control. All samples were incubated with 3 µM dimethyl sulfoxide (DMSO) for 48 h, as controls for experiments with corrector VX-809 (see Figure 3). Summary of data of CFTR variants on the wt- (**E**) or F508del- (**F**) CFTR backgrounds, expressed as normalized ratios (band C/(band C + band B)) and as a percentage to the respective ratio for wt-CFTR and as mean ± standard error of the mean (SEM). (n) indicates nr. of independent experiments. “*” and “#” indicate significantly different (*p* < 0.05) from wt-CFTR and F508del-CFTR, respectively.

**Figure 2 ijms-21-04524-f002:**
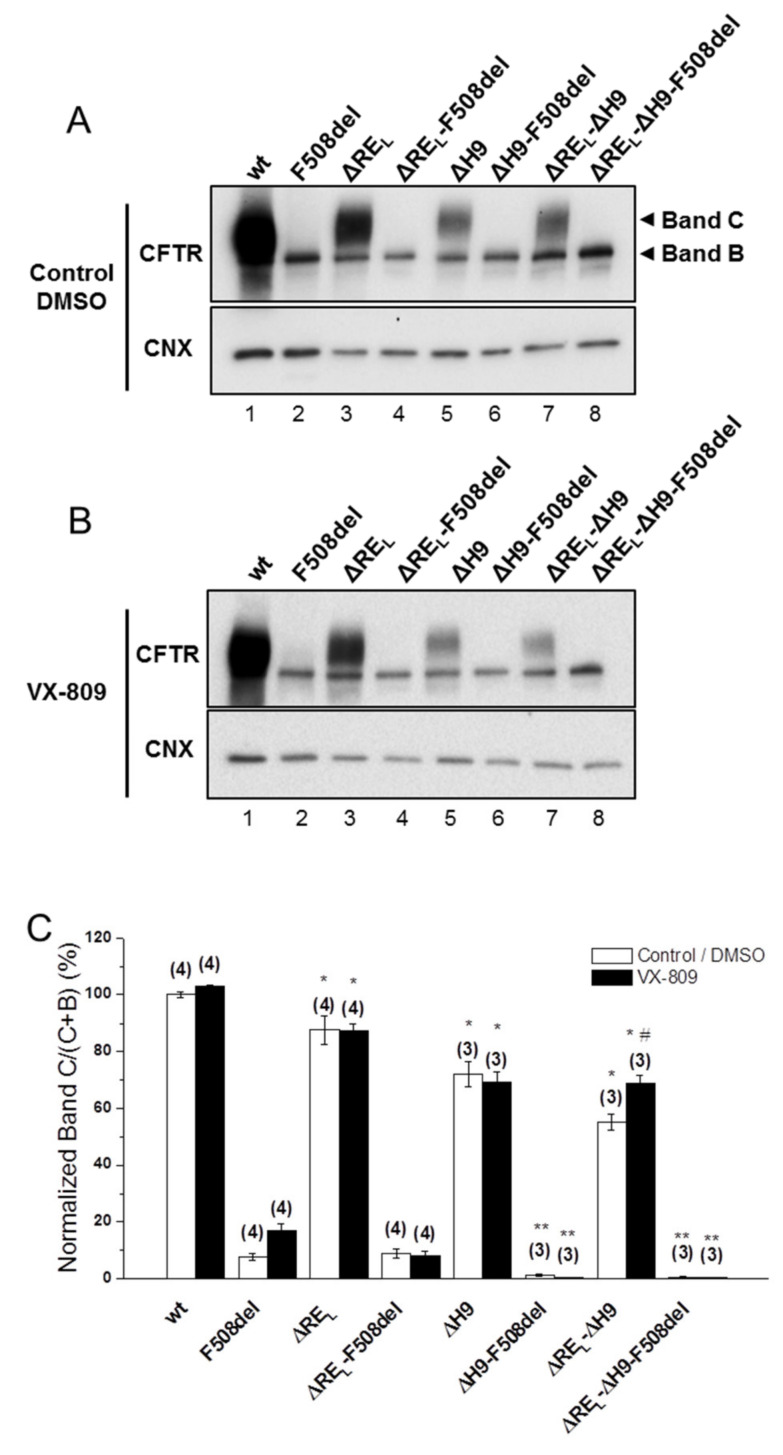
Effect of removal of helix H9 in combination with ΔRE_L_ and VX-809 on processing of wt- and F508del-CFTR. (**A**,**B**) WB analysis of samples from BHK cell lines stably expressing wt- or F508del-CFTR variants with ΔRE_L_, ΔH9, or ΔRE_L_-ΔH9. Cells were incubated with 3 µM VX-809 or DMSO (control) for 48 h at 37 °C. CFTR protein expression was analyzed by WB with the anti-CFTR 596 and anti-CNX Abs. (**C**) Summary of data expressed as normalized ratios (Band C / (Band C + B)) and as a percentage to the respective ratios on the wt- or F508del-CFTR backgrounds and shown as mean ± SEM. (n) indicates number of independent experiments. “#” indicates significantly different from the respective variant treated with DMSO. “*” and “**” indicate significantly different from wt- or F508del-CFTR, respectively.

**Figure 3 ijms-21-04524-f003:**
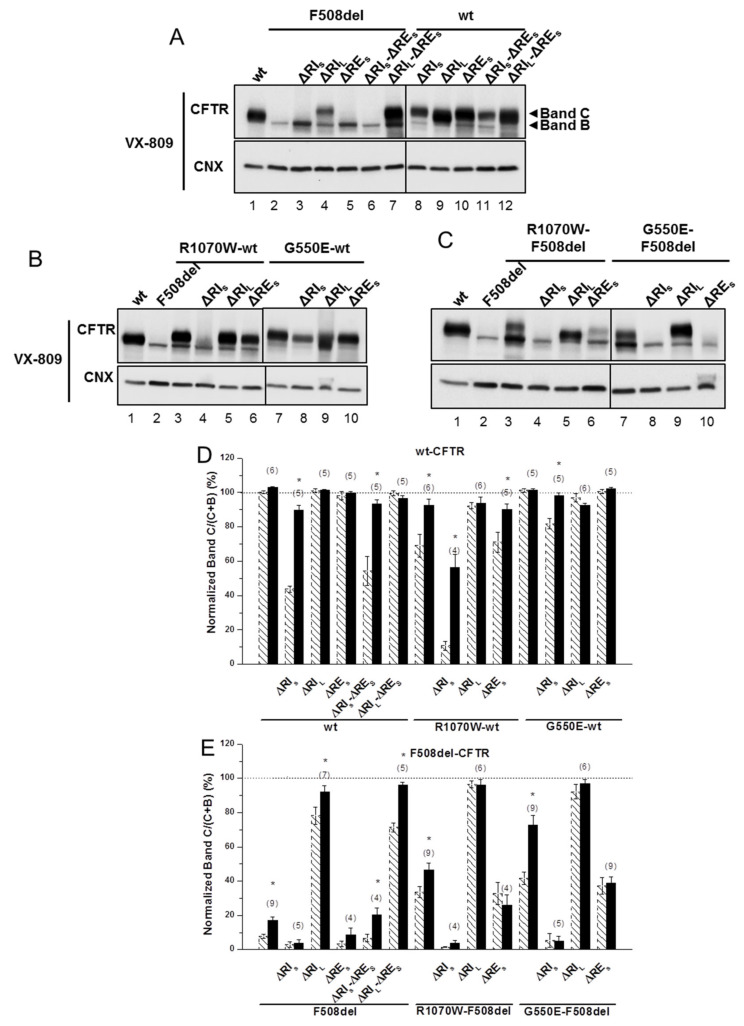
Effect of VX-809 on processing of wt- and F508del-CFTR variants without regulatory extension. (**A**–**C**) WB of samples from BHK cell lines stably expressing different CFTR variants of ΔRE and ΔRI, as indicated, alone (A) or jointly with (**B**,**C**) genetic revertants (see Materials and Methods). CFTR protein expression was analyzed by WB with anti-CFTR 596 and anti-CNX Abs as in Figure 1. All samples were incubated with 3 µM VX-809 for 48 h. (D, E) Summary of data of variants on the wt- (**D**) or F508del- (**E**) CFTR backgrounds, expressed as normalized ratios (band C/(band C + band B)) and as a percentage to the respective ratio for wt-CFTR and as mean ± SEM. (n) indicates number of independent experiments. “*” indicates significantly different (*p* < 0.05) from respective variant without VX-809 (as shown in Figure 1 and indicated here by dashed bars).

**Figure 4 ijms-21-04524-f004:**
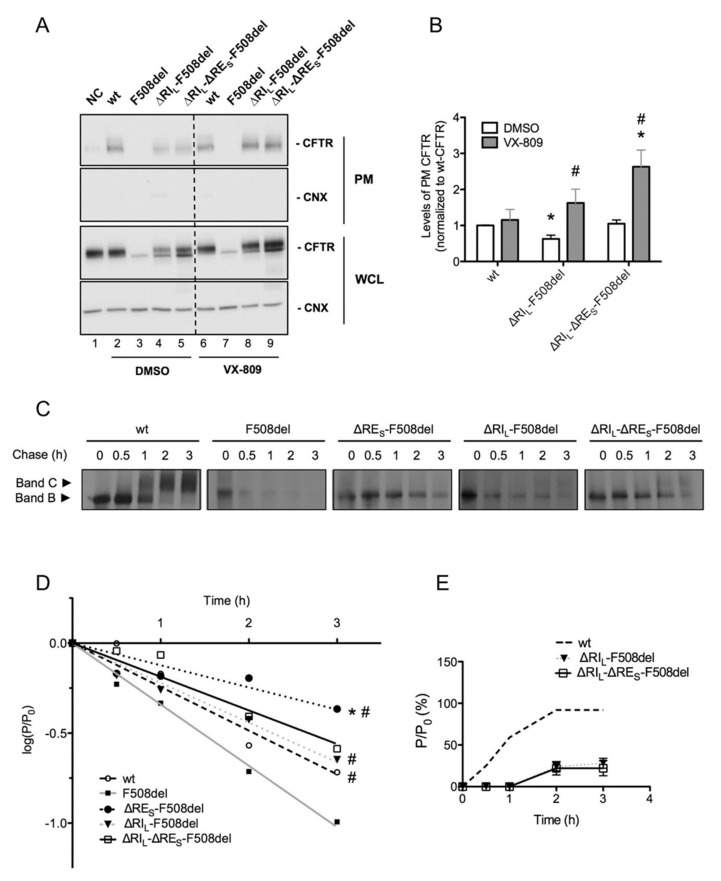
Plasma membrane levels, efficiency of processing and turnover of immature CFTR without Regulatory Extension (RE_S_). (**A**) BHK cells expressing ΔRI_L_-F508del and ΔRI_L_-ΔRE_S_-F508del-CFTR and treated with 3 µM VX-809 for 48 h (or DMSO control) were subjected to cell surface biotinylation. wt-CFTR samples not treated with biotin were the negative control (NC). After streptavidin pull-down, CFTR was detected by WB. CFTR and CNX are detected in the whole cell lysate (WCL) as controls. (**B**) Quantification of data in (a) for PM CFTR normalized to total protein and shown as fold change relatively to wt-CFTR cells treated with DMSO. “*” and “#” indicate significantly different from wt-CFTR treated with DMSO and from respective variant without VX-809, respectively (*p* < 0.05). (**C**) BHK cells expressing wt-, F508del-CFTR alone or jointly with ΔRI_L_, ΔRE_S_, and ΔRI_L_-ΔRE_S_ were subjected to pulse-chase (see Materials and Methods) for the indicated times (0, 0.5, 1, 2, and 3h) before lysis and immunoprecipitation (IP) with the anti-CFTR 596 Ab. After electrophoresis and fluorography, images were analyzed by densitometry. (**D**) Turnover of immature (band B) CFTR for different CFTR variants is shown as the percentage of immature protein at a given time point of chase (P) relative to the amount at t = 0 (P_0_). (**E**) Efficiency of processing of band B into band C is shown as the percentage of band C at a given time of chase relative to the amount of band B at t = 0. “*” and “#” indicate statistical significantly different (*p* < 0.05) from wt-CFTR and F508del-CFTR, respectively. Data represent mean ± SEM (*n* = 5).

**Figure 5 ijms-21-04524-f005:**
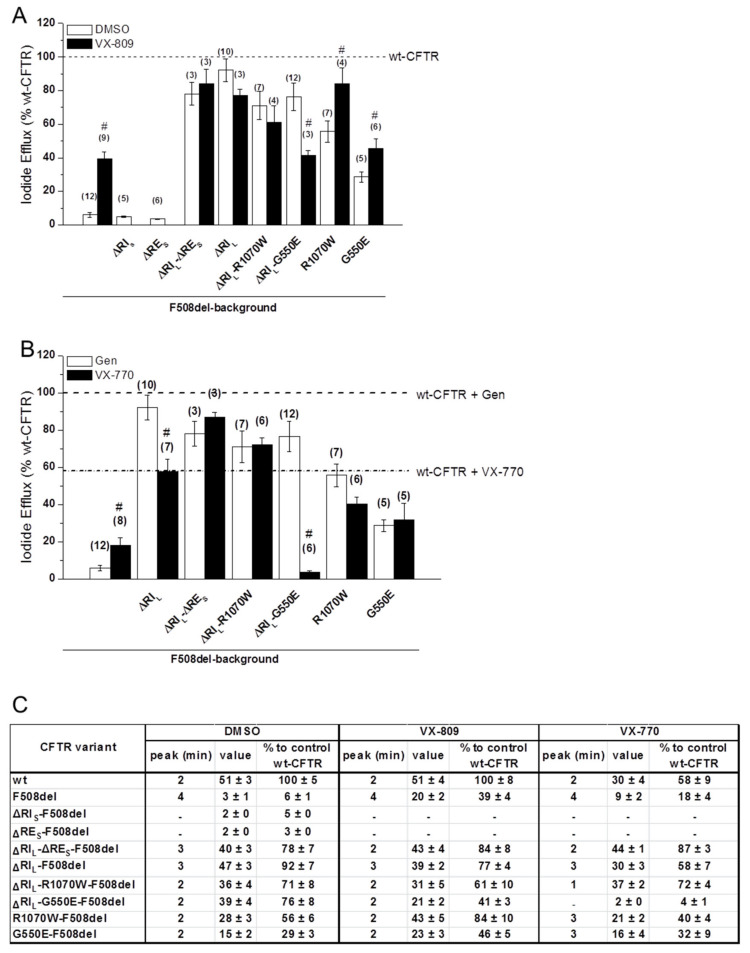
Functional characterization of the ΔRE_s_ and ΔRI_l_ variants of F508del-CFTR with or without VX-809 or VX-770 treatments. (**A**,**B**) Summary of the data from the iodide efflux peak magnitude generated by BHK cells stably expressing different CFTR variants and treated with 3 µM VX-809 (black bars on panel A) or with DMSO as control (white bars on panel a) for 48 h and stimulated with 10 µM Forskolin (Fsk) and 50 µM Genistein (Gen). In (**B**) cells were not pre-incubated and were stimulated either with 10 µM Fsk and 50 µM Gen (white bars on panel B) or with 10 µM Fsk and 10 µM VX-770 (black bars on panel b), as indicated. Data are shown as a percentage of wt-CFTR activity under Fsk/Gen stimulation and as mean ± SEM. (n) indicates number of independent experiments. “#” indicates significantly different (*p* < 0.05) from the same CFTR variant incubated with DMSO (A) or significantly different from the same CFTR variant under Fsk/Gen stimulation (white bars) (B). (**C**) Summary of iodide efflux data for different F508del-CFTR variants without RE_s_, RI_s_, and RI_l_ alone or jointly with revertants R1070W and G550E.

**Figure 6 ijms-21-04524-f006:**
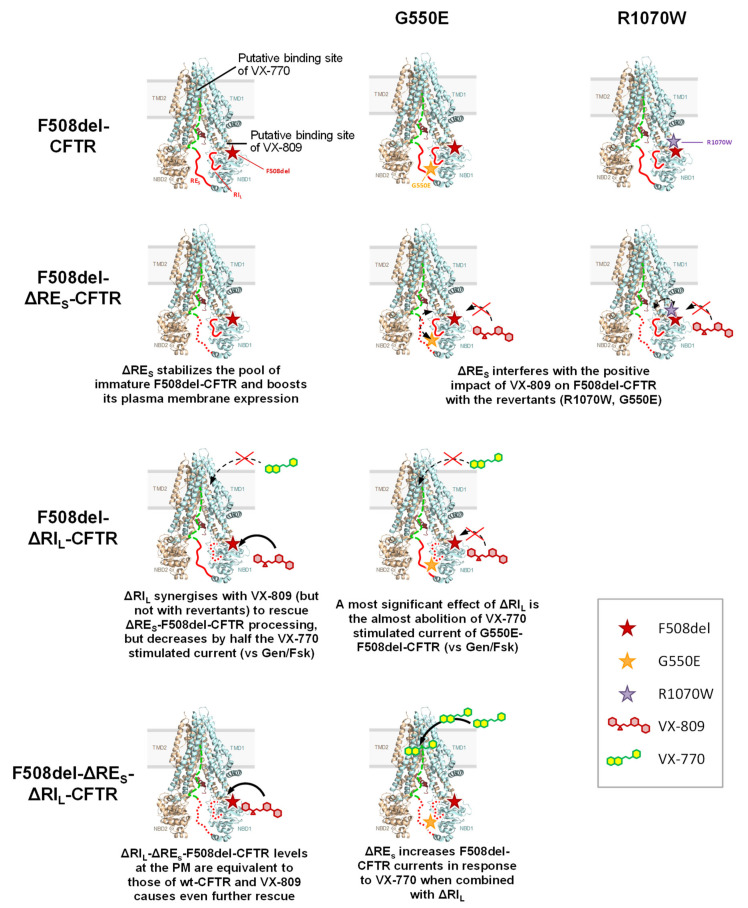
Summary of the most relevant results observed in the present study. CFTR structure used is the one by Liu F et al. [22] and putative bind sites of VX-809 and VX-770 shown are the NBD1:MSD2 (ICL4) interface and to the MSDs (although it is not defined the exact binding site), as described by Farinha et al. [5] and by Jih and Hwang [23], respectively. RI_L_ and RE_S_ are shown as red lines, the R region a dashed green line.

**Figure 7 ijms-21-04524-f007:**
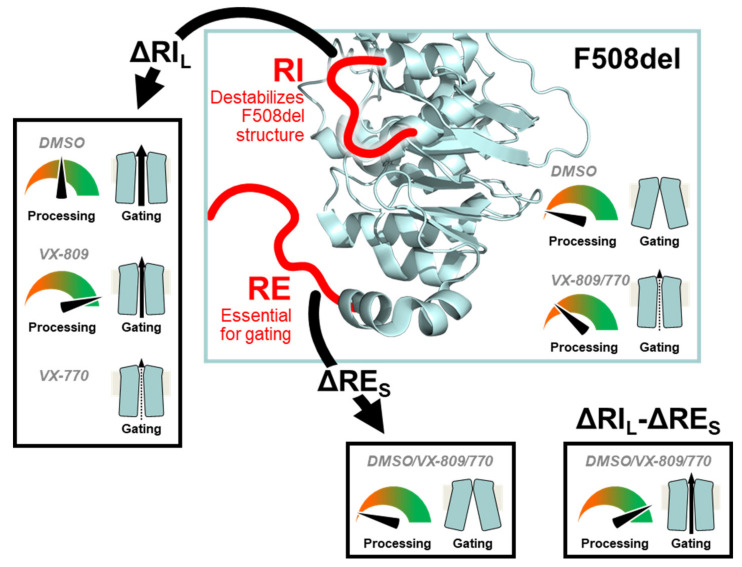
Model for the modulation of F508del processing and gating by RI_L_ and RE_S_. Our data suggest that the RI destabilizes the F508del-CFTR structure. Therefore, deleting the RI (ΔRI_L_) rescues F508del-CFTR processing and gating. However, the RI is essential for F508del-CFTR rescuing by VX-770. The RE is essential for gating and removal of its shorter version (ΔRE_S_) generates an F508del-CFTR variant which cannot be rescued pharmacologically. Simultaneous removal of both regions (ΔRI_L_-ΔRE_S_) rescues F508del-CFTR processing and gating. The CFTR structure from Liu F et al. [22] depicts the overall NBD1 structure and the location of the RI and RE.

**Table 1 ijms-21-04524-t001:** Summary of Western blot quantification of original data in Figure 1, Figure 2 and Figure 3 for different CFTR variants in the presence of VX-809 or DMSO control. Data are expressed as normalized ratios (band C/band (B + C)) for each variant and as a percentage to wt-CFTR ratio.

		wt-Background		F508del-Background	
		Control	VX-809	Control	VX-809
	CFTR Variant	% to wt	% to wt	% to wt	% to wt
	**-**	100 ± 1	103 ± 1	8 ± 1	17 ± 2
**ΔRE_S_**	**-**	98 ± 2	100 ± 1	3 ± 2	9 ± 4
	**ΔRI_S_**	54 ± 9	93 ± 3	7 ± 2	20 ± 4
	**ΔRI_L_**	100 ± 1	97 ± 2	71 ± 3	96 ± 2
	**R1070W**	71 ± 6	90 ± 3	33 ± 7	26 ± 6
	**G550E**	101 ± 1	102 ± 1	37 ± 5	39 ± 4
**ΔRE_L_**	**-**	88 ± 5	87 ± 3	9 ± 2	8 ± 2
	**ΔH9**	55 ± 3	69 ± 3	0 ± 0	0 ± 0
**ΔH9**		72 ± 4	69 ± 3	1 ± 0	0 ± 0
**ΔRI_S_**	**-**	44 ± 2	90 ± 3	3 ± 2	4 ± 3
	**R1070W**	11 ± 3	56 ± 8	2 ± 0	4 ± 2
	**G550E**	82 ± 3	98 ± 1	6 ± 4	5 ± 3
**ΔRI_L_**	**-**	101 ± 1	101 ± 0	78 ± 5	92 ± 4
	**R1070W**	92 ± 2	94 ± 3	96 ± 2	96 ± 3
	**G550E**	97 ± 2	92 ± 2	92 ± 4	97 ± 2
**R1070W**		69 ± 7	93 ± 4	34 ± 3	46 ± 4
**G550E**		101 ± 1	101 ± 1	42 ± 4	73 ± 6

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
