# Peer review of "Full Rescue of F508del-CFTR Processing and Function by CFTR Modulators Can Be Achieved by Removal of Two Regulatory Regions"

_ijms, 2020, doi:10.3390/ijms21124524_

Round 1

Reviewer 1 Report

The paper by Uliyakina and co-authors describes experiments aimed at addressing the role of two different conformationally dynamic regions (RI and RE) in CFTR to the processing and function of the protein in absence and presence of CFTR drugs and with and without the F508del mutation. Because of the complex network of interactions of the RI and RE, alluded nicely to one place in the discussion where the authors comment that the RE could take the place of the RI in CFTR variants lacking the RI, a number of different constructs were tested. The resulting work is a comprehensive investigation that presents compelling data showing that these two disordered regions affect WT and F508del differently (RI affects F508del-CFTR processing; RE affects stability and gating), and also affect the mode of drug action. Considering the increasingly demonstrated importance of disordered regions in membrane proteins, the work is very timely. I have some minor comments below.

  1. In order to determine the effects of different regions of the RI and RE, the authors made CFTR variants missing RIS, RIL, RES and REL, and Fig. S1 shows which residues are indicated by each variant. However, there was no description of why make two different RI and two different RE mutants - that the ΔRIs and ΔREs mutants remove only disordered regions and keep the residues that adopt helical structures in crystal structures of CFTR NBD1, while the ΔRIL and ΔREL mutants remove all RI and RE residues. The authors show differential effects of removing the RIS, RIL, RES and REL, alone and in combination, and thus this explanation is needed. The authors may also want to switch the colouring of the shading for REs (or RIS). Currently, the dark pink colours the H1b and H1c of the RI and the disordered part of the RE, whereas the light pink shading highlights the disordered part of the RI and the H9c helix of the RE.
  2. There is some discussion of residues comprising the RE and RI on page 12. However, there is an error in defining the residues of the RI (line 323). residue 405 is a Gly and the H1b and H1c helices are not from residues 405-436, as that is the full RI. Identification of H1b and H1c residues would help explain the difference between RIS and RIL.
  3. I am confused regarding the statement at line 38. It was my understanding that phosphorylated CFTR showed greater ATPase activity and gating compared to non-phosphorylated CFTR, suggesting that phosphorylation is not subsequent to ATP binding and hydrolysis.
  4. The statement on line 104, which reads "So ... removal of REs jointly with RIS or RIL had no effect on F508del-CFTR processing." should be rephrased to be clearer. It is true that processing of ΔRIs-F508del-CFTR and ΔRIs-ΔREs-F508del-CFTR are similar and that processing of ΔRIL-F508del-CFTR and ΔRIL-ΔREs-F508del-CFTR are also similar, but processing of ΔRIs-ΔREs-F508del-CFTR and ΔRIL-ΔREs-F508del-CFTR differ. While the sentence as written is correct, it would be helpful to the reader if it was more explicit.
  5. The authors provide a nice explanation of why removing the H9 helix, describing its contacts to the RE. However, one would assume that this helix also contacts NBD1, at least in some states, such as when the RE adopts different conformations like the one in reference 22 of this manuscript.
  6. Interestingly, the authors observed differences in the mode of action of VX-809 (lumacaftor) and VX-770 (ivacaftor) in that the RI is essential to processing of F508del-CFTR by VX-770 but is inhibitory to processing of F508del-CFTR by VX-809. The authors should discuss the VX-770 data in light of the 2019 structure of CFTR showing ivacaftor bound to the MSDs, which is far from the RI in the CFTR structure.

I also have some grammatical/style suggestions:

  1. A number of acronyms are defined in the manuscript and not used again or only used once after the definition (eg. ERQC, MoA). The authors may wish to remove these acronyms.
  2. "FDA-approved" should be changed to "FDA approval" in line 49.
  3. The sentence at lines 66-68 should be re-worded.
  4. The authors highlight bands B and C as being used to calculate normalized ratios, but none of the western blots in the main manuscript have band B and band C labeled. While the CFTR scientific community understands which is band B and band C, readers who do not study CFTR may not know which band is band B and which is band C. And explanation of these bands when discussing the processing of the various CFTR proteins and labelling band B and band C in Fig. 1 will be helpful.
  5. There is an extra bracket on line 97.
  6. There is a typo in line 304.
  7. The last sentence (line 333 and 334) should be re-worded.

Author Response

Comments and Suggestions for Authors

The paper by Uliyakina and co-authors describes experiments aimed at addressing the role of two different conformationally dynamic regions (RI and RE) in CFTR to the processing and function of the protein in absence and presence of CFTR drugs and with and without the F508del mutation. Because of the complex network of interactions of the RI and RE, alluded nicely to one place in the discussion where the authors comment that the RE could take the place of the RI in CFTR variants lacking the RI, a number of different constructs were tested. The resulting work is a comprehensive investigation that presents compelling data showing that these two disordered regions affect WT and F508del differently (RI affects F508del-CFTR processing; RE affects stability and gating), and also affect the mode of drug action. Considering the increasingly demonstrated importance of disordered regions in membrane proteins, the work is very timely. I have some minor comments below.

Our response

We are very grateful to the Reviewer for the generous appreciation of our work. The authors are also grateful to the reviewer for the helpful comments.

  1. In order to determine the effects of different regions of the RI and RE, the authors made CFTR variants missing RIS, RIL, RES and REL, and Fig. S1 shows which residues are indicated by each variant. However, there was no description of why make two different RI and two different RE mutants - that the ΔRIs and ΔREs mutants remove only disordered regions and keep the residues that adopt helical structures in crystal structures of CFTR NBD1, while the ΔRIL and ΔREL mutants remove all RI and RE residues. The authors show differential effects of removing the RIS, RIL, RES and REL, alone and in combination, and thus this explanation is needed. The authors may also want to switch the colouring of the shading for REs (or RIs). Currently, the dark pink colours the H1b and H1c of the RI and the disordered part of the RE, whereas the light pink shading highlights the disordered part of the RI and the H9c helix of the RE.

Our response

We agree with the reviewer regarding the need for an explanation for making two different RI and two different RE variants. Thus, we have now: i) included a sentence in the introduction regarding the controversy of these region (p.2, 4th parag); ii) a paragraph in the beginning of the results explain the structural considerations which prompted us to test different options (p.3, 1st parag); iii) a full discussion of the implications is given in the discussion (p.14, 3rd parag).

We have also changed the colours in Figs.S1.

  1. There is some discussion of residues comprising the RE and RI on page 12. However, there is an error in defining the residues of the RI (line 323). residue 405 is a Gly and the H1b and H1c helices are not from residues 405-436, as that is the full RI. Identification of H1b and H1c residues would help explain the difference between RIS and RIL.

Our response

We believe that the reviewer is not correct, Gly is 404 (not 405).

To possibly explain the different processing levels of the RIs and RIL F508del-CFTR variants we have included a sentence in the Discussion (p.17, 3rd parag).

  1. I am confused regarding the statement at line 38. It was my understanding that phosphorylated CFTR showed greater ATPase activity and gating compared to non-phosphorylated CFTR, suggesting that phosphorylation is not subsequent to ATP binding and hydrolysis.

Our response

We agree with the reviewer, this was a mistake. We have now corrected it (p.1, last parag).

  1. The statement on line 104, which reads "So ... removal of REs jointly with RIS or RIL had no effect on F508del-CFTR processing." should be rephrased to be clearer. It is true that processing of ΔRIs-F508del-CFTR and ΔRIs-ΔREs-F508del-CFTR are similar and that processing of ΔRIL-F508del-CFTR and ΔRIL-ΔREs-F508del-CFTR are also similar, but processing of ΔRIs-ΔREs-F508del-CFTR and ΔRIL-ΔREs-F508del-CFTR differ. While the sentence as written is correct, it would be helpful to the reader if it was more explicit.

Our response

We have added a sentence to make this particular result clearer (p. 5, 1st parag).

  1. The authors provide a nice explanation of why removing the H9 helix, describing its contacts to the RE. However, one would assume that this helix also contacts NBD1, at least in some states, such as when the RE adopts different conformations like the one in reference 22 of this manuscript.

Our response

We do not understand this comment, since H9 is actually part of NBD1. Nevertheless, we have added a sentence in the discussion to make this point more explicit (p.17, 3rd parag).

  1. Interestingly, the authors observed differences in the mode of action of VX-809 (lumacaftor) and VX-770 (ivacaftor) in that the RI is essential to processing of F508del-CFTR by VX-770 but is inhibitory to processing of F508del-CFTR by VX-809. The authors should discuss the VX-770 data in light of the 2019 structure of CFTR showing ivacaftor bound to the MSDs, which is far from the RI in the CFTR structure.

Our response

We believe that the reviewer meant ‘gating of F508del-CFTR by VX-770’ (and not ‘processing’). Nevertheless, we have added a sentence to discuss this in light of the 2019 structure of CFTR (p.17, 4th parag).

I also have some grammatical/style suggestions:

  1. A number of acronyms are defined in the manuscript and not used again or only used once after the definition (eg. ERQC, MoA). The authors may wish to remove these acronyms.

Our response

Done.

  1. "FDA-approved" should be changed to "FDA approval" in line 49.

Our response

Done.

  1. The sentence at lines 66-68 should be re-worded.

Our response

The sentence starting "Our data show that although F508del-CFTR without RE …" has now been rephrased for clarification but instead of being in the Introduction it has been moved to the Results (p.10, 1st parag).

  1. The authors highlight bands B and C as being used to calculate normalized ratios, but none of the western blots in the main manuscript have band B and band C labeled. While the CFTR scientific community understands which is band B and band C, readers who do not study CFTR may not know which band is band B and which is band C. And explanation of these bands when discussing the processing of the various CFTR proteins and labelling band B and band C in Fig. 1 will be helpful.

Our response

We have added to all WB figures band B and band C and an explanation for what they represent is now provided in the text (p.3, 1st parag).

  1. There is an extra bracket on line 97.

Our response

Removed.

  1. There is a typo in line 304.

Our response

Corrected.

  1. The last sentence (line 333 and 334) should be re-worded.

Our response

The sentence beginning "Interestingly however, removal of the RES from ΔRIL-F508del-CFTR …" has now been rephrased for clarification (p.15, 1st parag).

Reviewer 2 Report

The study in question addresses an important issue concerning the CFTR protein: its processing and opening.

CFTR is an ABC transporter containing two transmembrane domains forming a chloride ion channel, and two nucleotide binding domains, i.e. NBD1 and NBD2.

The leading cause of cystic fibrosis CF is the deletion of phenylalanine 508 (F508del) in the NBD1 of the CFTR. Two unique regions - regulatory extension (RE) and regulatory insertion (RI) - are present in NBD1 of CFTR, but they are absent elsewhere, i.e. in NBDs of other transporters.

NBD1 mutant fails to traffic to the plasma membrane due to protein misfolding and retention by the endoplasmic reticulum quality control that targets it to proteasomal degradation.

Several treatments can partially rescued F508del-CFTR folding.

Goals here were

  1. to assess the impact of removing the 30-amino acid RE (ΔRE) alone or jointly with ΔRI on the traffic of F508del-CFTR.
  2. to evaluate how RE and or RI removal from F508del-CFTR influences the rescue of this mutant by genetic revertants.
  3. to determine how the traffic and function of these combined variants of F508del-CFTR (ΔRE, ΔRI plus genetic revertants) are rescued by CFTR modulator drugs VX-809 (corrector) and VX-770 (potentiator) to gain further insight into their mechanism of action.

I believe the study is really well done. I congratulate the team of authors.

The objectives have been met by arriving at proposing that, despite RI and RE regions being conformationally active, RI precludes F508del-CFTR processing while RE affects mostly its stability and channel opening.

Data presentation is sufficiently clear and methods are appropriate.

Just one note: I suggest to report the Figure S2 (NBD1 structural details) - extremely explanatory thanks to the colors - between the main figures and not relegated to the Supplem Info.

Author Response

Reviewer #2

Comments and Suggestions for Authors

The study in question addresses an important issue concerning the CFTR protein: its processing and opening.

CFTR is an ABC transporter containing two transmembrane domains forming a chloride ion channel, and two nucleotide binding domains, i.e. NBD1 and NBD2.

The leading cause of cystic fibrosis CF is the deletion of phenylalanine 508 (F508del) in the NBD1 of the CFTR. Two unique regions - regulatory extension (RE) and regulatory insertion (RI) - are present in NBD1 of CFTR, but they are absent elsewhere, i.e. in NBDs of other transporters.

NBD1 mutant fails to traffic to the plasma membrane due to protein misfolding and retention by the endoplasmic reticulum quality control that targets it to proteasomal degradation.

Several treatments can partially rescued F508del-CFTR folding.

Goals here were

  1. to assess the impact of removing the 30-amino acid RE (ΔRE) alone or jointly with ΔRI on the traffic of F508del-CFTR.
  2. to evaluate how RE and or RI removal from F508del-CFTR influences the rescue of this mutant by genetic revertants.
  3. to determine how the traffic and function of these combined variants of F508del-CFTR (ΔRE, ΔRI plus genetic revertants) are rescued by CFTR modulator drugs VX-809 (corrector) and VX-770 (potentiator) to gain further insight into their mechanism of action.

I believe the study is really well done. I congratulate the team of authors.

The objectives have been met by arriving at proposing that, despite RI and RE regions being conformationally active, RI precludes F508del-CFTR processing while RE affects mostly its stability and channel opening.

Data presentation is sufficiently clear and methods are appropriate.

Our response

We are very grateful to the Reviewer for the generous appreciation of our work. The authors are also grateful to the reviewer for the helpful comments.

Just one note: I suggest to report the Figure S2 (NBD1 structural details) - extremely explanatory thanks to the colors - between the main figures and not relegated to the Supplem Info.

Our response

We consider this a very good suggestion and have included now Fig.S2 as panel A in Fig.1
